# Cardioprotective Effect of Novel Matrix Metalloproteinase Inhibitors

**DOI:** 10.3390/ijms21196990

**Published:** 2020-09-23

**Authors:** Kamilla Gömöri, Tamara Szabados, Éva Kenyeres, Judit Pipis, Imre Földesi, Andrea Siska, György Dormán, Péter Ferdinandy, Anikó Görbe, Péter Bencsik

**Affiliations:** 1Cardiovascular Research Group, Department of Pharmacology and Pharmacotherapy, Faculty of Medicine, University of Szeged, H-6720 Szeged, Hungary; gomori.kamilla@med.u-szeged.hu (K.G.); szabados.tamara@med.u-szeged.hu (T.S.); kenyeres.eva@med.u-szeged.hu (É.K.); gorbe.aniko@med.u-szeged.hu (A.G.); 2Pharmahungary Group, H-6722 Szeged, Hungary; judit.pipis@pharmahungary.com (J.P.); peter.ferdinandy@pharmahungary.com (P.F.); 3Department of Laboratory Medicine, Faculty of Medicine, University of Szeged, H-6720 Szeged, Hungary; foldesi.imre@med.u-szeged.hu (I.F.); siska.andrea@med.u-szeged.hu (A.S.); 4Targetex Biosciences, H-2120 Dunakeszi, Hungary; dorman@target-ex.com; 5Department of Pharmacology and Pharmacotherapy, Faculty of Medicine, Semmelweis University, H-1089 Budapest, Hungary

**Keywords:** matrix metalloproteinase inhibitor, thiazole and imidazole carboxylic acid derivatives, MMP-2, acute myocardial infarction, cardioprotection, cardiovascular comorbidity, hypercholesterolemia

## Abstract

Background: We recently developed novel matrix metalloproteinase-2 (MMP-2) inhibitor small molecules for cardioprotection against ischemia/reperfusion injury and validated their efficacy in ischemia/reperfusion injury in cardiac myocytes. The aim of the present study was to test our lead compounds for cardioprotection in vivo in a rat model of acute myocardial infarction (AMI) in the presence or absence of hypercholesterolemia, one of the major comorbidities affecting cardioprotection. Methods: Normocholesterolemic adult male *Wistar* rats were subjected to 30 min of coronary occlusion followed by 120 min of reperfusion to induce AMI. MMP inhibitors (MMPI)-1154 and -1260 at 0.3, 1, and 3 µmol/kg, MMPI-1248 at 1, 3, and 10 µmol/kg were administered at the 25th min of ischemia intravenously. In separate groups, hypercholesterolemia was induced by a 12-week diet (2% cholesterol, 0.25% cholic acid), then the rats were subjected to the same AMI protocol and single doses of the MMPIs that showed the most efficacy in normocholesterolemic animals were tested in the hypercholesterolemic animals. Infarct size/area at risk was assessed at the end of reperfusion in all groups by standard Evans blue and 2,3,5-triphenyltetrazolium chloride (TTC) staining, and myocardial microvascular obstruction (MVO) was determined by thioflavine-S staining. Results: MMPI-1154 at 1 µmol/kg, MMPI-1260 at 3 µmol/kg and ischemic preconditioning (IPC) as the positive control reduced infarct size significantly; however, this effect was not seen in hypercholesterolemic animals. MVO in hypercholesterolemic animals decreased by IPC only. Conclusions: This is the first demonstration that MMPI-1154 and MMPI-1260 showed a dose-dependent infarct size reduction in an in vivo rat AMI model; however, single doses that showed the most efficacy in normocholesterolemic animals were abolished by hypercholesterolemia. The further development of these promising cardioprotective MMPIs should be continued with different dose ranges in the study of hypercholesterolemia and other comorbidities.

## 1. Introduction

Acute myocardial infarction (AMI) is a major consequence of ischemic heart disease and is considered as the single most common cause of death, accounting for 1.8 million annual deaths or 20% of all deaths in Europe [1].

Since the discovery of endogenous cardioprotective mechanisms against ischemia/reperfusion (I/R) injury (local and remote ischemic pre-, post-, and perconditioning, pharmacological conditioning), several molecular contributors of cardioprotective maneuvers were explored. However, even after promising preclinical attempts aiming to trigger these cardioprotective mechanisms, the translation of the results into clinical practice has remained untried. The development of new cardioprotective compounds is challenging due to translation difficulties, including difficulties in reproducibility, at least in part due to the significant number of non-responder animals [2,3] and, moreover, due to the presence of several additional factors including cardiovascular co-morbidities, e.g., hyperlipidemia or diabetes mellitus [4]. Many cardioprotective strategies act through common end-effectors and may be suboptimal in patients with comorbidities [5]. Therefore, to improve clinical translation, novel therapeutic strategies are needed against myocardial I/R injury, which may preserve cardioprotection even in the presence of co-morbidities [6].

Hypercholesterolemia and metabolic disease are common comorbidities resulting from a sedentary lifestyle and an increased intake of saturated and unsaturated trans-fatty acids [7]. Elevated low-density lipoprotein (LDL) cholesterol causes endothelial and myocardial dysfunction, as well as exacerbating I/R-induced myocardial injury. Endogenous cardioprotective mechanisms against I/R have been shown to be impaired in a hyperlipidemic in vivo rat AMI model [8] and in vitro primary neonatal cardiomyocytes [9]. The cardioprotective effect of ischemic preconditioning (IPC) is lost in hypercholesterolemia, at least in part due to the peroxynitrite-induced activation of matrix metalloproteinase (MMP)-2 [10].

MMPs are zinc-dependent endopeptidases and the gelatinase-type MMP-2 isoenzyme is abundant in the heart tissue, synthesized by cardiac myocytes, fibroblasts, and endothelial cells. MMP-2 is synthesized as its inactive zymogen form and it can be activated via limited proteolysis [11] or conformational changes induced by reactive oxygen/nitrogen species [12,13]. MMP-2 has a critical role as an intracellular mediator of cardiac I/R injury contributing to the acute mechanical dysfunction that occurs immediately following reperfusion [14]. The main intracellular targets of MMP-2 in I/R injury are the proteins of the contractile apparatus [15], such as Troponin I [16], titin, myosin light chain-1 [17], and α-actinin in cardiac myocytes [18].

Because of the fundamental role of MMP-2 in I/R induced myocardial injury, the inhibition of MMP-2 seems a promising target in therapy for AMI. The first-generation MMP inhibitors were hydroxamic acid-type molecules, whose mechanism of action was based on Zn^2+^ ion chelation [19,20]. Ilomastat, a hydroxamic acid-type non-selective MMP inhibitor, was tested in preclinical studies. In ex vivo and in vivo mouse hearts, Bell et al. investigated the cardioprotective effect of ilomastat (0.25 µmol/L) administered upon reperfusion in an isolated in vitro model and also in an in vivo mouse model of ischemia/reperfusion injury (1.5 µmol/kg) and it attenuated infarct size equivalently to ischemic postconditioning, which was used as the positive control [21]. Our group previously showed its cardioprotective effect in ex vivo isolated rat hearts [10] and an in vivo acute myocardial infarction rat model [22]. Additionally, the mild inhibition of MMP-2 by ilomastat showed a comparable cardioprotection as IPC against I/R injury in the heart of normolipidemic rats and this protection was preserved even in the presence of hyperlipidemia [10].

MMP inhibitors have been developed for different indications, including cardiovascular diseases, but their translation into human clinical practice (except for periodontitis) has failed so far [23]. In 2006, in an early clinical trial (PREMIER), a hydroxamic acid-type MMP inhibitor (PG-116800) was tested in AMI patients [24,25]. Despite the success of several preclinical studies with this compound, the clinical trial failed to improve any major outcomes, probably due to insufficient target validation and the lack of optimal selectivity and affinity of the inhibitor towards MMP-2 [24]. The only exception was the hydroxamic acid derivative doxycycline [26], which showed a significant cardioprotective effect in AMI patients; however, the number of enrolled patients was limited [25]. Therefore, it is reasonable to consider the development of novel small-molecule structures for more selective and moderate inhibitors of MMP-2 with cardioprotective indication.

In our previous study on imidazole and thiazole carboxylic acid-based compounds, novel MMP inhibitor molecules were developed (Figure 1). These imidazole and thiazole carboxylic acid-based compounds showed superior MMP-inhibiting effects when compared to the hydroxamic acid-type derivatives in vitro in gelatin zymography assays. Six compounds, including MMP inhibitors (MMPI) 1154, MMPI-1260 and MMPI-1248 were found to be protective against simulated ischemia/reperfusion induced injury in cardiac myocytes [27]. Here, we further test these lead molecular compounds in vivo in relation to acute myocardial infarction in normo- and hypercholesterolemic animals, as hypercholesterolemia is one of the most frequent co-morbidities of acute myocardial infarction.

The aim of the present study was to achieve cardioprotection by using the most efficacious compounds (MMPI-1154, -1260, and -1248) in an in vivo rat model of AMI, in normocholesterolemic animals and in the presence of hypercholesterolemia.

## 2. Results

### 2.1. Testing the Cardioprotective Effects of Different Doses of MMP Inhibitor Compounds against AMI in Normocholesterolemic Rats

#### 2.1.1. Tolerability of MMP Inhibitors

In order to ascertain of the safety of the novel inhibitors, preliminary tolerance testing was performed with each inhibitor to determine their safety. Mean arterial blood pressure (Table 1), heart rate (Table 2) and body temperature were not changed throughout the experiments compared to the vehicle-treated animals. Vehicle-treated animals were treated with the appropriate volume of dimethyl sulfoxide (DMSO) at each treating timepoint (i.e., at 0; 20; 40; 60 and 80 min during the experiments).

There was no sign of organ failure. Only the urine was discolored and appeared dark; this phenomenon might have occurred due to the known hemolytic effect of dimethyl sulfoxide (DMSO) [28]; otherwise, rats tolerated the DMSO well [29]. Since DMSO is a strong organic solvent, we maximized the intravenous (iv.) volume at 60 µL for each animal; this amount dissolved the inhibitors completely and is a feasible volume to inject using a 1-mL syringe. No mortality occurred during the testing period in the case of MMPI-1260, only one animal died in the MMPI-1248 test group and two in the MMPI-1154 test group (Figure 2).

#### 2.1.2. All-Cause Mortality

All-cause mortality did not differ among experimental groups and the mortality rate was low. In most cases, mortality occurred before the administration of MMP inhibitors. For details, see Appendix A.

#### 2.1.3. Infarct Size

MMPI-1154, -1260, and -1248 inhibitors were shown to be cardiocytoprotective in in vitro cell culture experiments, as we previously published [27], and here we tested their efficacy in vivo.

After the 30-min coronary occlusion and 120-min reperfusion (for the experimental protocol, see Figure 3A) the area at risk data was determined (see Appendix A). The area at risk did not show any difference among experimental groups (Appendix A); therefore, the infarct size data are comparable. Two of the MMP inhibitor compounds showed a significant reduction in infarct size: MMPI-1154 at 1 µmol/kg and MMPI-1260 at 3 µmol/kg decreased infarct size significantly compared to the vehicle group (Figure 3B), from 63.68 ± 1.91% to 53.53 ± 3.36% and 56.64 ± 2.46%, respectively. MMPI-1248, in terms of infarct size reduction, did not exert cardioprotection in any of the applied doses when compared to the ischemic control group (Figure 3B); however, a decreasing tendency at 10 µmol/kg can be observed. Therefore, the two MMP inhibitor compounds exhibiting significant infarct size-lowering effects were further studied in hypercholesterolemic conditions. However, in spite of previous studies, which have demonstrated the infarct size-limiting effect of ilomastat in a rodent model of AMI [21,22], ilomastat failed to reduce infarct size (60.46 ± 3.60%) in the present study, showing the problems with the reproducibility of cardioprotective studies.

#### 2.1.4. Incidence of Severe Arrhythmias

Ventricular tachycardia (VT) and fibrillation (VF) were recorded during the whole surgical intervention. Left anterior descending (LAD) occlusion-induced ischemia had the possibility of triggering arrhythmia in the rat hearts. The MMP inhibitors, as well as ilomastat or the vehicle, were given at the 25th min of ischemia. Representative photos of the appearance of tachyarrhythmias are available in the Appendix A. There were no significant differences in the incidence of severe arrhythmias among experimental groups (Appendix A).

### 2.2. Testing the Cardioprotective Effects of Single Dose of MMP Inhibitor Compounds against AMI in Hypercholesterolemia (Hchol) and Age-Matched Normocholesterolemic (Nchol) Rats

#### 2.2.1. Hypercholesterolemic Model Validation

Male *Wistar* rats developed hypercholesterolemia due to a 12-week cholesterol-enriched diet. At the end of the 12-week diet, the induction of hypercholesterolemia was validated from baseline blood samples (after 12 h fasting) by measuring serum total cholesterol (Nchol: 2.10 ± 0.04 mmol/L and Hchol: 8.09 ± 0.38 mmol/L, respectively) and LDL levels (Nchol: 0.49 ± 0.03 mmol/L, and Hchol: 6.69 ± 0.34 mmol/L, respectively), which were significantly higher in the hypercholesterolemic group compared to the age-matched normocholesterolemic group; meanwhile, triglyceride levels (Nchol: 0.71 ± 0.04 mmol/L and Hchol: 0.55 ± 0.01 mmol/L, respectively) were significantly lower in the hypercholesterolemic group. HDL levels did not show any significant difference between groups (Appendix A). Hypercholesterolemic animals had significantly higher blood glucose levels (4.90 ± 0.23 mmol/L) compared to the age-matched normocholesterolemic group (3.67 ± 0.27 mmol/L) (Appendix A). However, body weights did not show any difference in the two major groups (Appendix A).

#### 2.2.2. All-Cause Mortality

Hypercholesterolemia did not affect all-cause mortality, which did not differ among experimental groups and the mortality rate was low (Appendix A). In most cases, mortality occurred before the administration of MMP inhibitors.

#### 2.2.3. Infarct Size

In the age-matched normocholesterolemic group, IPC significantly reduced infarct size (26.23 ± 6.16%) compared to the vehicle-treated ischemic group (55.58 ± 3.41%) and both MMP inhibitor molecules provided cardioprotection (MMPI-1154: 40.61 ± 3.38% and MMPI-1260: 36.75 ± 4.63%) reproducibly in terms of the infarct size reduction (Figure 4B) compared to the results of the age-matched normocholesterolemic model (Figure 3B). Area at risk data showed no difference among groups (Appendix A). In the presence of hypercholesterolemia (Figure 4C), IPC (36.77 ± 6.6%), MMPI-1154 at 1 µmol/kg (44.82 ± 6.1%) and MMPI-1260 at 3 µmol/kg (44.03 ± 2.4%) failed to reduce infarct size when compared to the vehicle-treated control group (45.59 ± 4.8%).

#### 2.2.4. MMP-2 and MMP-9 Activities

In order to test baseline MMP-2 and MMP-9 levels, the activities of enzymes were tested in plasma samples of age-matched normocholesterolemic and hypercholesterolemic animals. The baseline MMP-2 and MMP-9 activity in hypercholesterolemic animals in comparison to age-matched normocholesterolemic animals were not different (Figure 5).

#### 2.2.5. Microvascular Obstruction

Ischemia/reperfusion causes damage to cardiomyocytes and also to the coronary microcirculation, including the impairment of endothelial integrity with subsequently increased permeability and edema formation, platelet activation and leukocyte adherence and, ultimately, structural damage to the capillaries with an eventual no-reflow phenomenon, which is called microvascular obstruction (MVO; see representative photo in Figure 6A).

Both in the age-matched normocholesterolemic and hypercholesterolemic groups, the positive control, IPC, significantly reduced MVO (Nchol: 4.45 ± 0.75%, Hchol: 5.35 ± 1.68%) compared to the vehicle-treated ischemic group (Nchol: 9.67 ± 1.47%, Hchol: 12.57 ± 2.70%). The tested MMP inhibitor molecules did not change MVO compared to the vehicle-treated ischemic group (Nchol MMPI-1154: 7.49 ± 1.56% and MMPI-1260: 9.19 ± 2.23%; Hchol MMPI-1154: 11.09 ± 2.34% and MMPI-1260: 13.45 ± 2.07%); see Figure 6B,C.

#### 2.2.6. Incidence of Severe Arrhythmias

Ventricular tachycardia (VT) and fibrillation (VF) were monitored during the ischemic period in the hypercholesterolemic model as well. Similarly, in relation to the normocholesterolemic model, there were no significant differences among the groups (Appendix A).

### 2.3. Hemodynamics

There was no significant difference in mean arterial blood pressure or heart rate among the groups, either in the normocholesterolemic (Appendix AA–C) or hypercholesterolemic (Appendix A) models. The mean arterial blood pressure dropped at the beginning of the ischemia as a result of coronary occlusion. The altered function adjusted to the injury at the end of the ischemic period with gradual depletion in the reperfusion phase (Appendix A). In terms of the heart rate, there was no significant difference during the time course of measurement among experimental groups (Appendix A).

## 3. Discussion

In the present study, we show the dose-dependent cardioprotective effect of novel MMP inhibitors MMPI-1154 and -1260 in an in vivo rat model of AMI when administered before reperfusion. However, in the presence of hypercholesterolemia, their infarct size-limiting effect was not seen in a single dose that showed cardioprotective effects in normal rats. This is the first demonstration that MMPI-1154 and MMPI-1260 are cardioprotective in vivo against myocardial infarction. Whether hypercholesterolemia inhibits their cardioprotective effect or may only shift the dose–response relationship of these compounds remains unknown.

An effective cardioprotective drug for the treatment of ischemic heart disease is still an unmet clinical need since several promising drug candidates failed to protect the ischemic myocardium in large clinical studies. Among several cardioprotective drug targets, the participation of MMP-2 in the development of reperfusion injury following extended myocardial ischemia was clearly demonstrated in the early 2000s. Schulz and colleagues showed that MMP-2 is activated and thereby degrades myocardial contractile proteins such as myosin light chain [30], titin [31] and troponins [16], as well as sarcoplasmic/ endoplasmic reticulum Ca^2+^ ATPase 2a (SERCA2a) [32] and junctophilin-2 [33] (see [34,35]) during myocardial ischemia/reperfusion injury. Simultaneously, the therapeutic potential has arisen to inhibit MMP-2 activity in order to prevent the degradation of the abovementioned cardiac proteins. In isolated, perfused rat hearts after global ischemia, the acute release of MMP-2 during reperfusion contributed to cardiac mechanical dysfunction and it was improved by the administration of MMP inhibitors such as doxycycline (10 to 100 mmol/L) and o-phenanthroline (3 to 100 mmol/L) [14]. Recently, a clinical trial [25] was conducted in which doxycycline marginally reduced infarct size (*p* = 0.052) and significantly attenuated myocardial remodeling.

Despite several decades of investigation to develop cardioprotective agents against AMI, there is still a significant need for new cardioprotective drug candidates. Therefore, MMP inhibitor development became a novel potential strategy of cardiovascular drug development.

The first types of MMP inhibitors were hydroxamic acid-based molecules, which bind to the catalytic zinc ion of MMPs, resulting in poor selectivity among MMP isoforms [20]. A limitation of the prototype matrix metalloproteinase inhibitor marimastat, which was developed to combat malignant tumors, was—beyond poor efficacy—the occurrence of musculoskeletal syndrome [36]. Non-selective MMP inhibitors were tested as potential cardioprotective compounds in several preclinical studies. Coronary flow and heart rate were improved by *o*-phenanthroline (100 µM) in isolated rat hearts subjected to I/R injury [37].

According to Jacobsen et al., to achieve highly specific MMP inhibitors, hydroxamate and carboxylate zinc-binding groups could be developed by careful consideration of inhibitor backbones for targeting the different substrate pockets of individual MMPs [20]. Our attempts to increase the selectivity for MMP-2 against MMP-1 and other MMPs were embedded in the pyridine moiety instead of the phenyl ring at the end of the S1′ pocket, occupying a longer side chain of the novel inhibitor candidates (MMPI-1260, 1248) [27]. Another important aspect during inhibitor screening was to inhibit MMP-2 activity moderately, which may be accompanied by limited side effects [35,38]. Our molecules were developed through a screening cascade from the in silico Albany Molecular Ressearch Inc. (AMRI) Chemical Library to find zinc-binding motif-holding molecules similar to hydroxamic acids. Thiazole and isosteric imidazole carboxylic acids were chosen and screened through different in vitro assays in order to select the most potent MMP inhibitors for in vivo experiments aiming for cardioprotection (for details, see [27]).

In the present study, three selected imidazole (MMPI-1154) and thiazole (MMPI-1248 and -1260) carboxylic acid-type MMP-2 inhibitors were tested in an in vivo rat model of AMI. MMPI-1154, -1260 and -1248 were tested in three different doses based on previous half maximal inhibitory concentration (IC_50_) values (IC_50_ for MMP-2 MMPI-1154: 2.5 µM, MMPI-1260: 2.6 µM and MMPI-1248: 9 µM) [27]. MMPI-1154 at 1 µmol/kg and MMPI-1260 at 3 µmol/kg showed cardioprotection by significantly lowering infarct size compared to the vehicle-treated group in normocholesterolemic rats subjected to AMI. Only a few studies have been performed in in vivo models of acute myocardial infarction to investigate the cardioprotective effect of MMP inhibitors. One group was using a 48-h daily pretreatment of doxycycline in rats prior to 30 min of coronary occlusion followed by 2 days of reperfusion. Doxycycline pretreatment significantly reduced infarct size compared to the untreated group [39]. In another study, rats were administered a vehicle or minocycline, a semi-synthetic tetracycline-type MMP inhibitor, via intraperitoneal injection at 25 mg/kg every 12 h for 48 h before and after 45 min of LAD occlusion. After 48 h of reperfusion, infarct size was significantly reduced by minocycline treatment [40]. ARP-100 (2-[([1,1’-biphenyl]-4-ylsulfonyl)(1-methylethoxy)amino]-N-hydroxy-acetamide), a biphenylsulfonamide-type MMP inhibitor is one of the most selective small-molecule MMP-2 inhibitors over other MMP inhibitors available on the market so far (IC_50_ for MMP-2 = 12 nM, MMP-9 = 200 nM, and MMP-3 = 4500 nM) [32]. ARP-100 shows selectivity towards MMP-2 and MMP-9 [41]. A recent study on isolated rat hearts demonstrated the cardioprotective effect of ARP-100 (10 mM) against myocardial stunning when administered 10 min prior to 20-min myocardial ischemia [33].

Hypercholesterolemia is a common comorbidity associated with cardiovascular diseases [4], which severely interferes with endogenous cardioprotective conditioning [8]. Therefore, investigating the role of MMP inhibition in acute myocardial infarction in the presence of hypercholesterolemia has significant importance. We have previously shown that MMP-2 activity correlates positively with serum total (*r* = 0.55; *p* < 0.05) and LDL cholesterol levels (*r* = 0.45; *p* < 0.05) in coronary artery disease patients [42]. In the present study, hypercholesterolemia was induced by a 12-week high-fat diet (2% cholesterol + 0.25% cholic acid). We tested the cardioprotective doses of MMP-1154 and -1260 in hypercholesterolemic rats, which showed cardioprotection in normocholesterolemic animals in this study. MMPI-1154 (1 µmol/kg) and MMPI-1260 (3 µmol/kg), administered i.v. 5 min before reperfusion, showed significant cardioprotection reproducibly in normocholesterolemic age-matched animals but not in hypercholesterolemic ones. In our previous study on MMP-2 inhibition by ilomastat, we showed a reduced infarct size in both normo- and hyperlipidemic hearts. However, that study differed in three major aspects compared to the present study: it used (i) isolated rat hearts derived from animals fed 2%-enriched cholesterol for 12 weeks; (ii) cholic acid was not used, and ilomastat was perfused into the hearts continuously; (iii) this started 45 min before the onset of global ischemia and was maintained during the entire reperfusion [10]. Previously, we showed that ilomastat can decrease myocardial infarct size in normocholesterolemic rats in vivo when administered either before ischemia or, at a higher dose (6.0 µmol/kg), administered before the onset of reperfusion [22]. Others found similar effects of ilomastat in mice [21]. However, controversially, in the present study, ilomastat (6.0 µmol/kg administered 5 min before reperfusion) failed to reduce infarct size in vivo, but our novel MMP inhibitors were cardioprotective, reproducibly. In another study, using isolated rabbit hearts, doxycycline treatment led to the inhibition of MMP-2 activity and a reduction in infarct size in normocholesterolemic and hypercholesterolemic animals [43]. However, others have found that doxycycline was unable to prevent disease progression on calcified aortic valve in apoE−/−mice [44].

In this study, we used ilomastat as a positive control to diminish reperfusion injury after AMI. It is now clearly demonstrated that ischemic postconditioning may lack cardioprotective effects both in patients as well as in preclinical animal models (see [45]). In a recent study, we highlighted the presence of a phenomenon that accompanies ischemic postconditioning and leads to diverse effects, distinguishing responders and non-responders to postconditioning stimuli in terms of the infarct size reduction. This phenomenon is supposed to be, at least partially, based on the individual polymorphisms of the downstream signaling pathways of postconditioning stimuli, including Jun dimerization protein type 2 (JDP2) and activator protein-1 (AP-1) [2]. Therefore, in the present study, we may suppose that the ratio of animals not responding to the cardioprotective signals evoked by ilomastat was higher than previously published, which might lead to the loss of significant infarct size reductions. Additionally, hypercholesterolemia could disguise the protective effect of these novel MMP inhibitors, since it was demonstrated that the altered metabolism due to hypercholesterolemia leads to a better tolerance of ischemia/reperfusion injury and thus to a smaller infarct size [46]. In this study, the infarct size data derived from hypercholesterolemic rats were significantly lower than that of in normocholesterolemic animals (see Appendix A), a difference that could feasibly abolish the significant infarct size-limiting effect of our novel MMP inhibitors. In light of the above factors influencing infarct size data, although we could not show the significant infarct size-lowering effect of our novel MMP inhibitors in hypercholesterolemic conditions, their cardioprotective potential cannot be excluded since we tested them in only in single doses. Testing further doses of MMPI-1154 and -1260 may show their cardioprotective capacity even in the presence of hypercholesterolemia. Furthermore, our novel MMP inhibitors may possess late beneficial effects on cardiac function in post-MI heart failure models.

Microvascular obstruction (MVO) is a phenomenon that refers to the observation of no reflow after the ischemic insult of certain organs. In experimental models, no-reflow zones in the heart show ultrastructural microvascular damage, including endothelial swelling obstructing the lumen of small vessels [47]. In ST elevation myocardial infarction (STEMI) patients who undergo a successful primary percutaneous intervention, coronary microvascular dysfunction and obstruction occurs in every second person, and MVO is associated with a much worse outcome [48]. To rescue the heart from microvascular obstruction after AMI and to avoid microembolization, cardioprotective strategies are waiting to be established in clinical practice [49]. Therefore, it is important to investigate MVO in preclinical studies, to gather information for a better understanding of the pathomechanism and to find possible therapeutic options. In a large animal model, late gadolinium-enhanced cardiac MRI was used to assess the myocardial MVO burden after different ischemic conditionings [50]. So far, several findings have shown that myocardial edema and MVO are altered independently of myocardial necrosis, which is in line with the findings of clinical trials [50]. In the present model, we used thioflavine-S staining perfused through the whole heart and the heart slices were investigated under UV light to visualize the penetration of the dye into the tissue. Similar to the abovementioned pig model [50], our results also show a significantly reduced MVO in (both normo- and hypercholesterolemic) preconditioned groups compared to the ischemic control group. However, none of the MMP inhibitors could reduce the percentage of MVO, either in age-matched normocholesterolemic or in hypercholesterolemic animals. This may imply that our novel MMP inhibitors may achieve cardioprotection that does not affect the coronary microvasculature.

### 3.1. Future Perspectives

More attention needs to be paid to the proper design of future clinical trials, in particular to the experimental power and dosage. The design of improved, safer, and more specific MMP inhibitors needs to continue. A key issue is that no drug has yet been shown or specifically designed to inhibit intracellular MMP-2 or its isoforms. To achieve clinical success, drugs must be innovative in either how they inhibit MMPs, how selectivity is achieved, how they are administered to the patient, or a combination of these attributes [23].

In a recent paper, Davidson et al. discussed several trials investigating cardioprotection against ischemia/reperfusion injury that failed. They suggest that new strategies for cardioprotective attempts, such as multitargeted therapy, might be the future, where cardioprotective agents are combined with interventions and non-cardiac myocyte targets, leading to an improvement in microcirculatory flow after AMI [5]. According to this new concept, MMP inhibition does have a rationale in the future care of AMI.

### 3.2. Limitations of the Present Study

In this study, we treated rats subjected to myocardial ischemia/reperfusion injury at the 25th min of coronary occlusion (5 min before the onset of reperfusion) with a single bolus dose. We based the dose range of all compounds on the previous findings in isolated neonatal cardiac myocytes [27]. However, the examination of the pharmacokinetic or full absorption, distribution, metabolism, and excretion and toxicology (ADMETox) properties of the inhibitor compounds exceeded the scope of the present work.

In the experiments with normocholesterolemic animals, we used ilomastat as a positive control to decrease myocardial infarct size after I/R injury. However, in spite of previous studies, which have demonstrated the infarct size-limiting ability of ilomastat in rodent models of AMI [21,22], this time it was unable to achieve cardioprotection in terms of infarct size reduction compared to the vehicle control. Therefore, due to the inefficacy of ilomastat to reduce infarct size in normocholesterolemic rats, we changed the positive control to IPC in the hypercholesterolemic model, despite the fact that its clinical relevance and applicability is limited. We also provided a detailed discussion above about the potential reasons of the absence of the cardioprotective effect of ilomastat with this administration algorithm.

Furthermore, we found that hypercholesterolemia interferes with myocardial infarct size; thereby, the infarct size-limiting effect of our novel drug candidates could be disguised by hypercholesterolemia.

## 4. Materials and Methods

### 4.1. Animals

The present study conforms to the EU directive about the care and use of laboratory animals, published by the European Union (2010/63/EU) and it was approved by the National Scientific Ethical Committee on Animal Experimentation (approval ID: XXVIII./171/2018.; on 24 January 2018) and by the Ethics Committee for Animal Research of the University of Szeged. Animals were housed in individually ventilated cages (Sealsafe IVC system, Tecniplast S.p.a., Varese, Italy), which conform the size recommendations of the abovementioned EU guidelines. Litter material (Lignocell hygienic animal bedding) placed beneath the cage was changed at least three times a week. The animal room was temperature controlled (22 ± 2 °C), and it had a 12-h light/dark cycle. The animals were acclimatized in the housing facility for 5 days prior to the start of the animal experiments. Animals were fed with standard rodent chow and filtered tap water was available ad libitum. Animals from the hypercholesterolemic groups were fed with standard rodent chow supplemented with 2% cholesterol (04820, Molar Chemicals, Halásztelek, Hungary) and 0.25% cholic acid (C1254, Sigma Aldrich, St. Louis, MO, USA).

### 4.2. Experimental Model of Acute Myocardial Infarction

All surgical procedures were performed as described previously [51] and modified according to the present study (Figure 3A and Figure 4A). Male *Wistar* rats weighing 260–340 g were anesthetized by ip. injection of pentobarbital sodium (Repose 50%, Le Vet. Pharma, Oudewater, The Netherlands). The rats were weighed and their stomach and chest area shaved. Maintenance of the body core temperature (37 ± 1 °C) was assisted using a constant temperature heating pad. The trachea was intubated with a plastic cannula connected to a rodent ventilator (Model 7025, Ugo Basile SRL, Gemonio, Italy). The animals were ventilated with room air (6.2 mL/kg, 70 ± 5 breath/min). Blood pressure, surface-lead electrocardiogram (ECG), and body core temperature were monitored throughout the experiments to ensure the stability of the preparation (Haemosys data acquisition system, Experimetria, Budapest, Hungary). The right carotid artery was cannulated for measurement of blood pressure. The right jugular vein was cannulated for the administration of the test compounds or vehicle. A thoracotomy was performed at the 5th intercostal space and the heart was exposed through the fifth intercostal space. A 5-0 Prolene suture was placed around the left anterior descending (LAD) coronary artery. The coronary artery was then occluded for 30 min by placing the ligature through a small piece of plastic tubing and pulling the snare tightly in place using a hemostat. After 30 min of coronary occlusion, reperfusion was initiated by releasing the snare and continued for 120 min.

### 4.3. Tolerability Testing

Before the use of novel inhibitors in AMI model, preliminary tolerance testing was performed with each molecule. For each safety test, *n* = 8 male *Wistar* rats were used. After anesthesia, animals were placed on a heating pad and rectal temperature, heart rate and mean arterial blood pressure (MABP) were monitored. Rats were given 5 increasing doses of the inhibitors dissolved in dimethyl sulfoxide (DMSO; D8418, Sigma-Aldrich, St. Louis, MO, USA). Distilled water, physiological saline, ethanol and DMSO were tested as potential vehicles. DMSO was chosen as the vehicle after testing the solubility of the molecules. Every 20 min, a 60-µL volume was administered iv. at 0.1, 0.3, 1.0, 3.0 and 10 µmol/kg. Twenty minutes after the administration of the highest dose, animals were terminated.

### 4.4. Animal Models and Treatments

Three-month-old animals were randomly assigned to the experimental groups with a total number of 15 animals/group. We performed two substudies to show the cardioprotective effects of the selected MMP inhibitor compounds in the normocholesterolemic and hypercholesterolemic models. In the normocholesterolemic model, three MMP inhibitor compounds (MMPI-1154, -1260, and -1248, see Figure 1) were tested against AMI in normocholesterolemic rats. Animals in the 1154 and 1260 test groups received 0.3, 1 and 3 µmol/kg of each MMP inhibitor. MMPI-1248 was administered in 1 µmol/kg, 3 µmol/kg or 10 µmol/kg amounts based on previous IC_50_ measurements [27]. MMP inhibitor compounds were injected intravenously (iv.) at the 25th min of ischemia, in slow bolus (through the right jugular vein). The vehicle (dimethyl sulfoxide; DMSO, D8418, Sigma-Aldrich)-treated group served as the negative control and DMSO was administered iv. in 60-µL amounts as a slow bolus. In the first experimental setup, in the normocholesterolemic model, we used ilomastat, a non-selective, hydroxamic acid-type MMP inhibitor, as a positive control to decrease myocardial infarct size after ischemia/reperfusion injury. Ilomastat was administered iv. at the 25th min of ischemia at 6 µmol/kg dose and repeated at the 10th and 25th min of reperfusion with half doses of 3 µmol/kg (according to [22]; also see Figure 3A).

In the hypercholesterolemic models, animals were divided into 2 major groups: (i) age-matched normocholesterolemic animals were fed with normal rodent chow for 12 weeks, while (ii) hypercholesterolemic animals received standard rodent chow supplemented with 2% cholesterol and 0.25% cholic acid. MMPI-1154 and -1260, in one efficacious dose for each (MMPI-1154 at 1 µmol/kg, MMPI-1260 at 3 µmol/kg), were tested against AMI in both groups. The animals were 6 months old at the time of surgery. The same (iii) vehicle group (DMSO) was used in the hypercholesterolemic model, as described above for both major groups. However, in the hypercholesterolemic model, IPC, the only known reproducibly efficacious cardioprotective maneuver was used as the positive control in the rat AMI model. Before the test ischemia, 3 cycles of 3-min ischemia and 5-min reperfusion were applied as the preconditioning stimuli (Figure 4A).

### 4.5. Determination of Myocardial Infarct Size

At the end of the 120-min reperfusion period, the heart was isolated and infarct size was determined as described previously [52]. Briefly, the LAD was re-occluded and the heart was perfused with 4 mL of 0.25% (*w*/*v*) Evans blue dye (E2129, Sigma-Aldrich, St. Louis, MO, USA) in Langendorff mode with a constant 100-cmH_2_O pressure into the aorta to delineate the area at risk. Stained hearts were rapidly frozen (−20 °C for at least 2 h), cut into 2-mm-thick slices (total of six), and each slice was incubated at 37 °C in 1 mL of 1% (*w*/*v*) 2,3,5-triphenyltetrazolium chloride (TTC, 108380, Merck Biosciences, Darmstadt, Germany) dissolved in 50 mM of phosphate buffer (pH 7.4) for 10 min. TTC penetrated the cardiac myocytes on the surface and, in living cells, TTC was reduced into a brick-red compound, while in the dead cells the stain remained colorless. Slices were then transferred to 10% formalin solution for 10 min, rinsed, and then placed between glass plates; finally, digital photos were taken (Canon, SX60 HS) from both sides of the heart slices. The differently stained areas of the heart images (white: infarcted region; red: area at risk; blue: non-ischemic region, see Appendix A) were quantified by digital planimetry (Infarctsize™ 2.5, Pharmahungary 2000 Ltd.). An evaluation of all images was carried out in a blinded manner by an experienced person throughout the study.

### 4.6. Microvascular Obstruction

In the hypercholesterolemic model (Figure 4A), microvascular obstruction [53] was measured from the isolated hearts. At the end of reperfusion, hearts were isolated and, before the reocclusion of LAD, thioflavine-S (T1892, Sigma-Aldrich, St. Louis, MO, USA) stain was perfused through the whole heart using the Langendorff retrograde perfusion system. Then hearts were stained with Evans blue, as described above in detail. The hearts were freshly cut into six 2-mm-thick slices and were placed in a dark chamber under UV light. The fluorescence of thioflavine-S was visible where it could penetrate the tissue through the coronary capillaries, except the areas where microvascular obstruction occurred (see representative images at Figure 6A).

### 4.7. Gelatin Zymography

In order to investigate MMP-2 and MMP-9 activity [54] from rat plasma samples, 50 µg of protein/lane was loaded and separated by electrophoresis on a 10% SDS-polyacrylamide gel copolymerized with 2 mg/mL gelatin from porcine skin (G1890, CAS 9000-70-8, Sigma-Aldrich; St. Louis, MO, USA). After electrophoresis, gels were washed in 2.5% TritonX 100 with gentle agitation and then incubated for 20 h at 37 °C in zymography development buffer (50 mM Tris-HCl, pH 7.5, containing 5 mM CaCl_2_, 200 mM NaCl). Zymographic gels were stained with 0.05% Coomassie Brilliant Blue G-250 dye (20279, CAS 6104-58-1, ThermoScientfic, Rockford, IL, USA), followed by destaining, then zymograms were scanned. MMP activity was detected as a colorless, transparent zone on a blue background (Appendix A) and the clear bands in the gel were quantified by densitometry using the Quantity One software (Bio-Rad, Hercules, CA, USA). The obtained density values are expressed in arbitrary units.

### 4.8. Lipid Panel Measurement

In the hypercholesterolemic model, to determine the development of hypercholesterolemia [55] after the 12-week cholesterol-enriched diet (Figure 4A), blood samples were collected from the animals before coronary occlusion surgery. Animals had 12 h of fasting prior to baseline blood sampling (cannulated carotid artery). Blood was sampled into heparinized tubes and centrifuged at 4 °C for 15 min at 1000× *g* to gain plasma. Total cholesterol, triglyceride, low-density lipoprotein (LDL), intermediate-density lipoprotein (IDL), high-density lipoprotein (HDL) and glucose levels were measured using a Hitachi Cobas8000 automated system at the Department of Central Laboratory Medicine of the University of Szeged.

### 4.9. Statistical Analysis

Data were expressed as the mean ± SEM. All-cause mortality and the incidence of arrhythmias were analyzed using a chi-square test. Area at risk, infarct size and the MVO of test compounds were compared to the vehicle using one-way ANOVA followed by Fisher’s LSD post-hoc test. The hemodynamic data of the test compounds were compared to the vehicle by repeated measures ANOVA followed by Fisher’s LSD post-hoc test. Lipid panel measurement compared the 12-week diet between normo- and hypercholesterolemic animals with Student’s *t*-test. A significance value of * *p* < 0.05 was chosen.

## 5. Conclusions

MMPI-1154 and MMPI-1260, shown to be cardiocytoprotective in vitro and ex vivo, have been further proved here to be cardioprotective in vivo when administered before the onset of reperfusion, which is a clinically relevant therapeutic approach in rat models of AMI. Although the presence of hypercholesterolemia abolished the cardioprotective effects of MMPI-1154 and -1260 in single doses that showed cardiprotection in vivo, whether this was due to a shift in the dose–response relationship of these compounds remains unknown. Further development of these promising cardioprotective MMPIs should be continued with different dose ranges in the study of hypercholesterolemia and other comorbidities.

## 6. Patents

MMP inhibitors (MMP-1154, MMPI-1260 and MMPI-1248) are novel chemical entities protected by patents (WO_2012/080762_A1, owned by Pharmahungary and TargetEx).

## Figures and Tables

**Figure 1 ijms-21-06990-f001:**
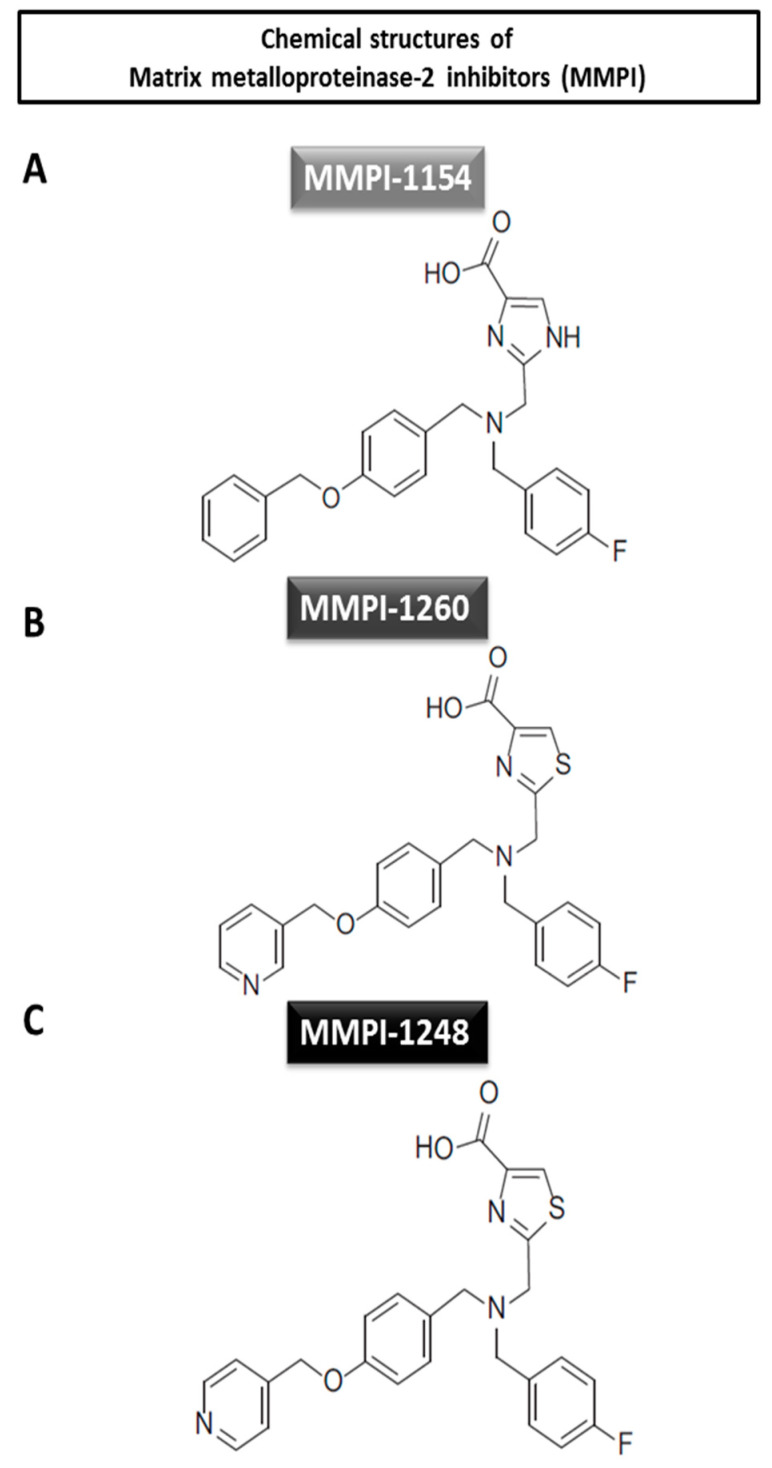
Chemical structures of the tested matrix metalloproteinase-2 inhibitors. (**A**) Imidazole-4-carboxylic acid derivate and (**B** and **C**) thiazole-4-carboxylic acid derivates.

**Figure 2 ijms-21-06990-f002:**
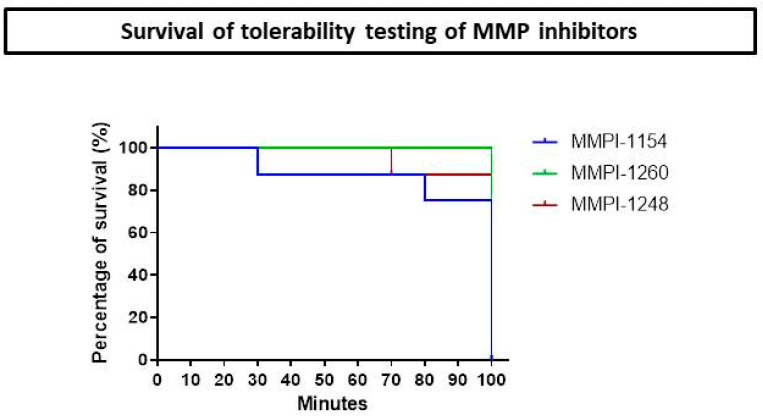
Kaplan-Meier curve of survival of male *Wistar* rats during matrix metalloproteinase inhibitor (MMPI) tolerability testing (*n* = 8).

**Figure 3 ijms-21-06990-f003:**
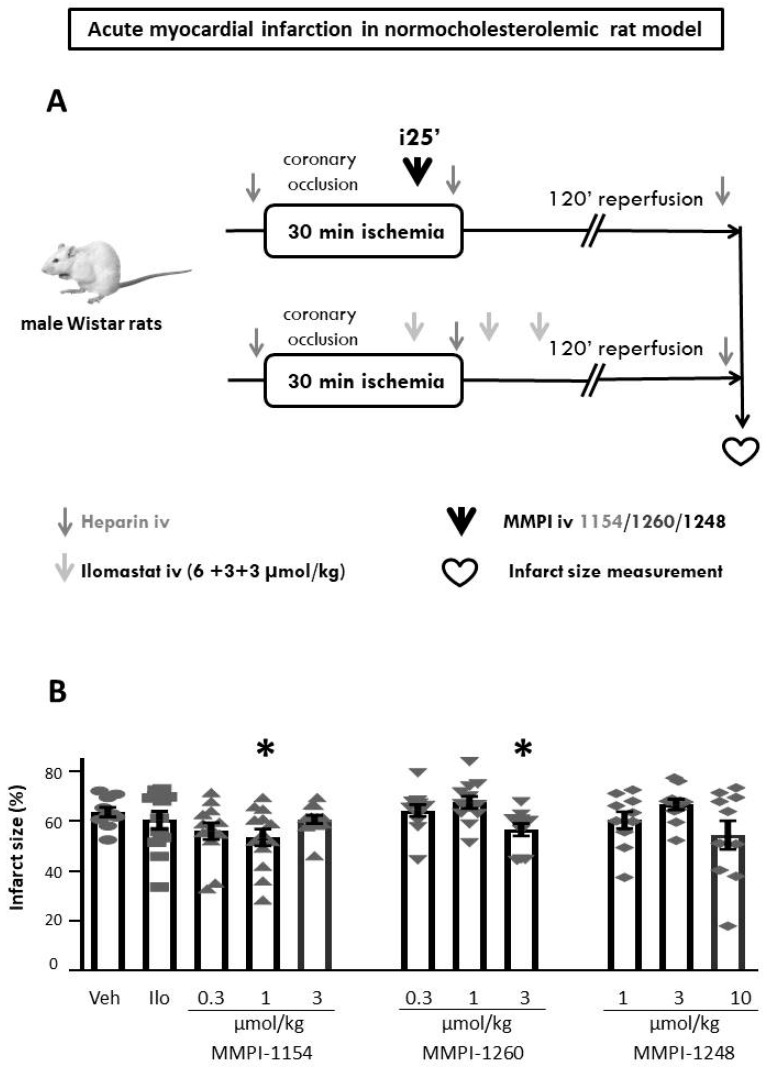
Experimental protocol of acute myocardial infarction. (**A**) in normocholesterolemic rat model. (**B**) The effects of MMP inhibitors on infarct size in normocholesterolemic rats subjected to in vivo 30-min coronary occlusion followed by 120-min reperfusion. Veh: dimethyl sulfoxide, Ilo: ilomastat. One-way ANOVA followed by Fisher’s Least Significant Difference (LSD) post-hoc test, *n* = 12–14, data are expressed as means ± SEM, * *p* < 0.05.

**Figure 4 ijms-21-06990-f004:**
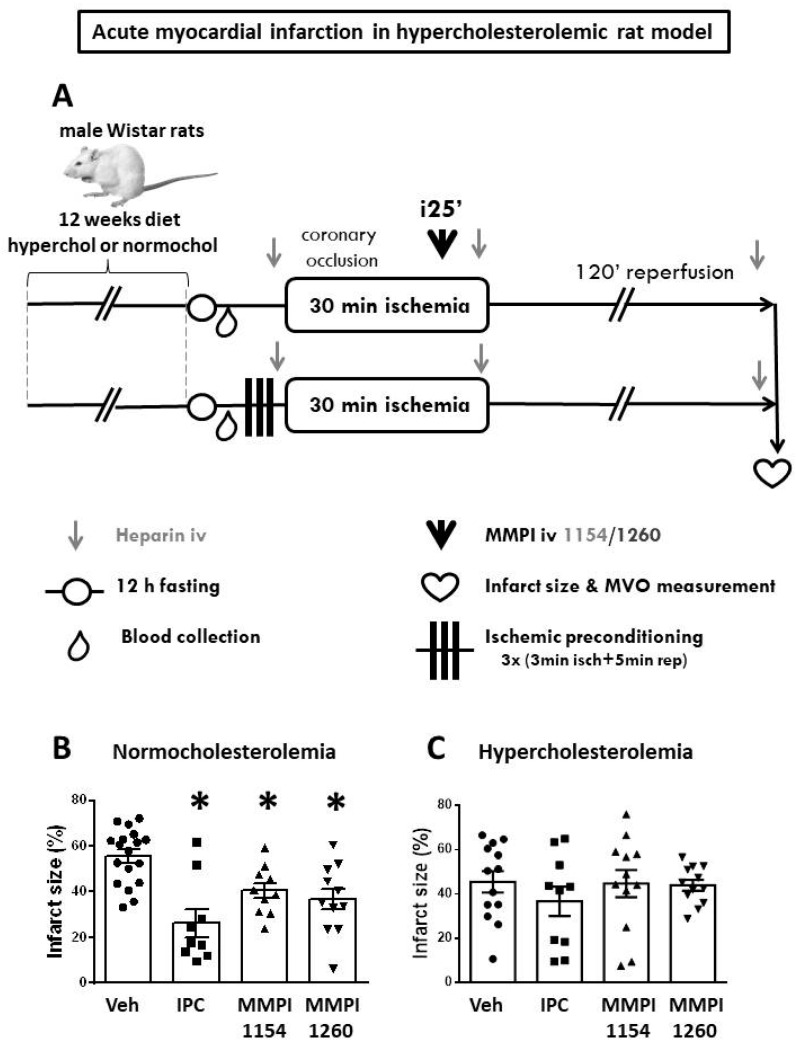
Hypercholesterolemic model. (**A**) In vivo experimental protocol of acute myocardial infarction in hypercholesterolemic rat model. The effects of ischemic preconditioning and MMP inhibitors on infarct size in normocholesterolemic (**B**) and hypercholesterolemic (**C**) rats subjected to in vivo 30-min coronary occlusion followed by 120-min reperfusion. Veh: dimethyl sulfoxide, IPC: ischemic preconditioning. One-way ANOVA followed by Fisher’s LSD post-hoc test, *n* = 12–14, data are expressed as means ± SEM, * *p* < 0.05.

**Figure 5 ijms-21-06990-f005:**
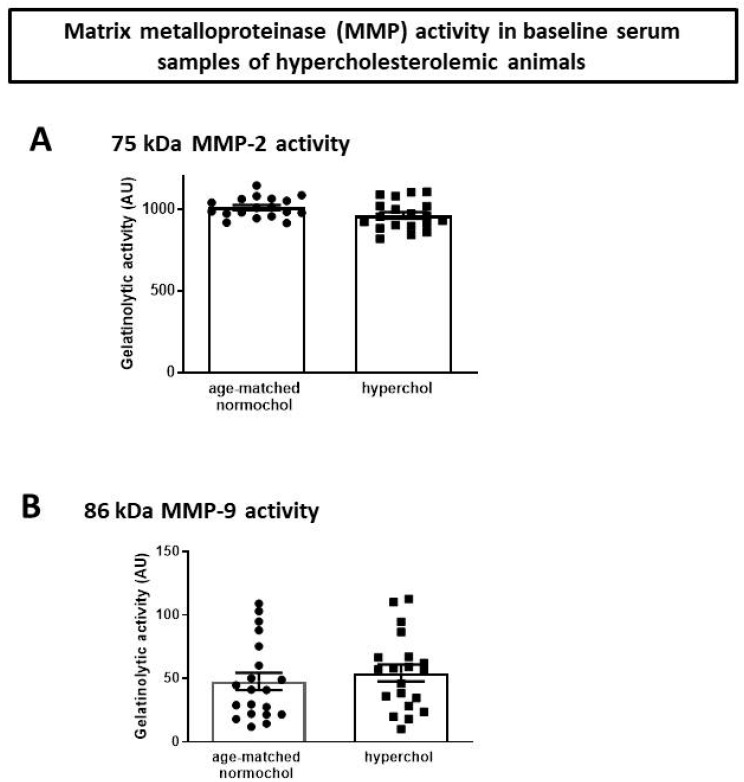
Gelatinolytic activity of (**A**) MMP-2 and (**B**) MMP-9. AU: arbitrary units.

**Figure 6 ijms-21-06990-f006:**
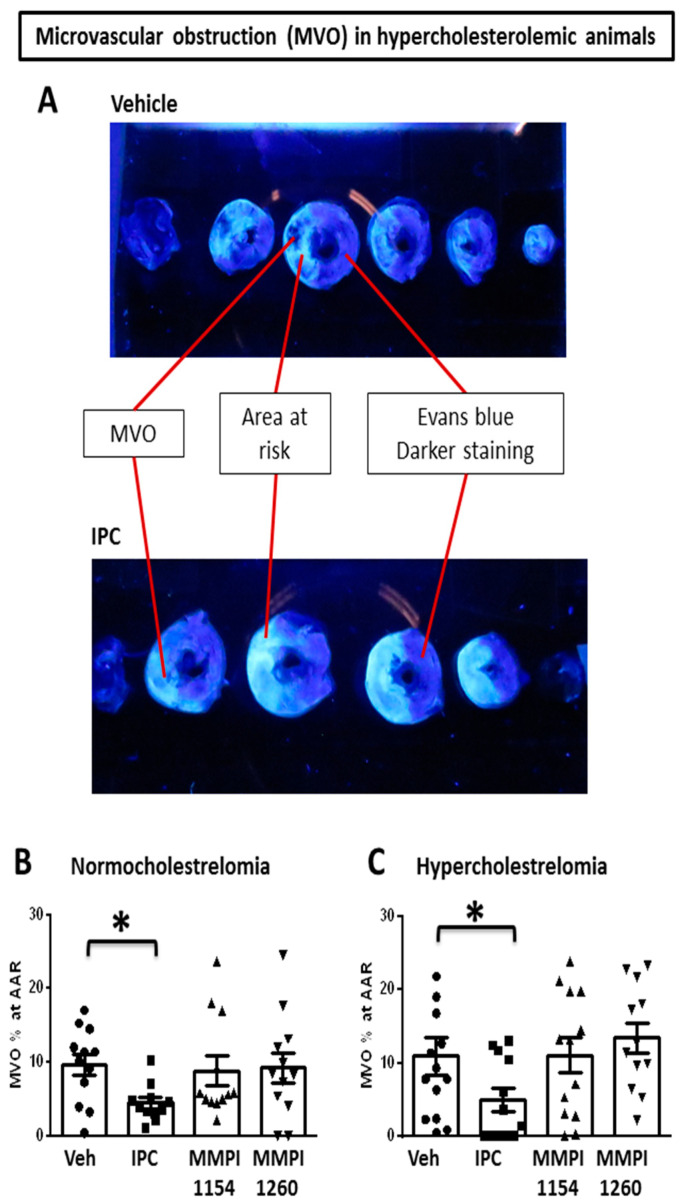
Microvascular obstruction (MVO) in rat heart after myocardial ischemia/reperfusion injury. (**A**) Representative image of MVO in Vehicle-treated group and IPC group. The effects of IPC and MMP inhibitors on MVO in (**B**) age-matched normocholesterolemic and (**C**) hypercholesterolemic rats subjected to in vivo 30-min coronary occlusion followed by 120 min reperfusion. Veh: dimethyl sulfoxide, IPC: ischemic preconditioning. One-way ANOVA followed by Fisher’s LSD post-hoc test, *n* = 14–16, data are expressed as means ± SEM, * *p* < 0.05.

**Table 1 ijms-21-06990-t001:** Mean arterial blood pressure (mmHg) of the animals during tolerability testing.

		0.1 µmol/kg		0.3 µmol/kg		1 µmol/kg		3 µmol/kg		10 µmol/kg		
MABP	Baseline	0′	10′	20′	30′	40′	50′	60′	70′	80′	90′	100′
Vehicle	132 ± 10	129 ± 15	120 ± 12	113 ± 13	113 ± 11	120 ± 12	112 ± 13	117 ± 11	107 ± 11	110 ± 10	113 ± 9	113 ± 7
MMPI-1154	151 ± 8	146 ± 7	133 ± 7	134 ± 8	138 ± 6	133 ± 8	140 ± 7	137 ± 7	124 ± 12	128 ± 5	134 ± 6	137 ± 7
MMPI-1260	138 ± 8	137 ± 8	121 ± 13	120 ± 18	123 ± 14	114 ± 15	122 ± 11	119 ± 12	116 ± 11	123 ± 10	126 ± 10	129 ± 10
MMPI-1248	105 ± 8	98 ± 8	103 ± 9	99 ± 8	90 ± 9	82 ± 7	88 ± 13	91 ± 10	89 ± 14	79 ± 11	110 ± 11	99 ± 9

Mean arterial blood pressure (mmHg) of the animals during tolerability testing. There was no significant difference in mean arterial blood pressure between groups compared to the vehicle-treated group, as analyzed by repeated measures two-way ANOVA, *n* = 8/group.

**Table 2 ijms-21-06990-t002:** Heart rate (beat/minute) of the animals during the tolerability testing.

		0.1 µmol/kg		0.3 µmol/kg		1 µmol/kg		3 µmol/kg		10 µmol/kg		
HR	Baseline	0′	10′	20′	30′	40′	50′	60′	70′	80′	90′	100′
Vehicle	425 ± 15	412 ± 14	418 ± 6	389 ± 9	405 ± 8	397 ± 10	379 ± 9	384 ± 6	387 ± 12	378 ± 8	376 ± 16	385 ± 17
MMPI-1154	436 ± 16	444 ± 12	403 ± 18	411 ± 19	411 ± 8	438 ± 10	393 ± 15	413 ± 14	389 ± 15	405 ± 17	390 ± 9	406 ± 15
MMPI-1260	435 ± 5	439 ± 4	422 ± 8	432 ± 15	417 ± 12	428 ± 6	413 ± 8	416 ± 12	411 ± 13	420 ± 12	413 ± 10	416 ± 10
MMPI-1248	440 ± 9	446 ± 11	417 ± 11	427 ± 17	400 ± 13	380 ± 21	386 ± 14	403 ± 18	377 ± 32	373 ± 19	384 ± 11	385 ± 19

Heart rate (beat/minute) of the animals during the tolerability testing. There was no significant difference in heart rate between groups compared to the vehicle-treated group, as analyzed by repeated measures two-way ANOVA, *n* = 8/group.

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
