# Peer review of "Cardioprotective Effect of Novel Matrix Metalloproteinase Inhibitors"

_ijms, 2020, doi:10.3390/ijms21196990_

Round 1

Reviewer 1 Report

In my opinion, the revision submitted by the authors addressed all the comments and suggestions of the reviewer. Thus, the manuscript can be accepted. Thank you. 

Reviewer 2 Report

Dear authors,

thank you for answering all my questions, and indeed, I now understand some of the outcomes of your experiments much better than before. I hope, your MMP(2)-inhibitors continue to be successful in patients!

This manuscript is a resubmission of an earlier submission. The following is a list of the peer review reports and author responses from that submission.

Round 1

Reviewer 1 Report

In this manuscript, the authors focused on 2 MMP2 inhibitors in order to demonstrate their cardioprotection effects against inschemia/reperfusion (I/R) in rats. The authors show how the inhibitors are able to reduce the infarct size in wild type rats after I/R injury, instead, when hypercholesterolemia is present, inhibitors fail to induce the beneficial effects found in normal conditions.

Please, find below the major and minor comments.

Major:

  • The data not shown in section “2.1.1. Tolerability of MMP inhibitors”, in my opinion, is of critical interest since the authors should be showing that the inhibitor is not toxic for the animals. Moreover, the authors should specify the experiments performed.
  • In my opinion, it would be a good idea to show MMP-2 levels: 1) in order to show that the treatment is actually working at the molecular level and 2) because maybe during hypercholesterolemia there is a modulation of the protein leading to the non-functional action of the inhibitor.
  • Moreover, I think it would be also interesting to see how the myocardial proteins (mentioned in the discussion) are modulated in controls and glycemic mice before and after treatment with the inhibitors.
  • The title says that the inhibitors have cardioprotective effects with or without hypercholesterolemia. Instead, in the results and conclusion (line 213-215) the authors state that the beneficial effects are abolished by hypercholesterolemia.

Minor:

  • In my opinion, there is no need for so many sub-sections (such as 2.1.X). Also because each sub-section is really short. Probably is better to just merge everything in the general section (such as 2.X).
  • In line 117, which data are the authors referring to when they say “data did not show any difference”. If it is the infarct size, please specify it within the explanation.
  • In the text, Supp. Fig. 3 is mentioned before Supp. Fig. 2. I would change the order.  
  • In the text, Fig 2B is also mentioned before Fig 2A. Figure 2A can be mentioned within the text while explaining the scheme/timings of the experiment.
  • Line 178, please correct “leucocyte” by “leukocyte”
  • English should be improved in order to help the text flow better.
  • Poor quality of the images.

Reviewer 2 Report

Dear Authors,

the manuscript surely addresses an important issue, that would greatly benefit many people upon success: The development of small pharmacological molecules, MMP2-inhibitors in this case, for cardioprotection in relation to myocardial infarction. You, among others, obviously succeeded in the experimental development of selective MMP2-inhibitors over the last years – as is extensively described in the introduction and discussion of this manuscript. Now, the (pre)clinical effectiveness of MMPIs has to be proven.

The experimental procedures are well described and the key experiment is nicely visualized by the protocol-graphics. The evaluation method (infarct size) is likewise explained using a graphic in the supplement.

Since the cardioprotective effect (reduction of infarct size) of MMP-Is seems to be rather small, the results of the preclinical experiments have to be evaluated very critically and stated clearly. The AMI model procedure in rats is surely challenging, and seemingly marginal results can be highly relevant, or at least interesting. But also vice versa. Thus, ambiguous or varying results have to be carefully discussed and explained to the reader. The critical evaluation and discussion of the results obtained in this study is mostly missing, except as a hint in the last section of “3.2 Limitations of the present study”.

And when experimental problems occur, then experiments have to be repeated as soon as the problems are cleared.

Main points:

1.) The title is misleading because it suggests that the novel MMP2 inhibitors mediate the cardioprotective effect in both normo- and hypercholesterolemic rat models. You should at least change it into something like “Novel MMP inhibitors mediate cardioprotection against myocardial infarction in normo- but not hypercholesterolemic rats” if this indeed is your conclusion. Or, if you want to be more cautious, you could just add at the beginning of the title “ Evaluation of novel mmp inhibitors for cardioprotection….”

2.) As you state yourself, it is strange that the MMP-inhibitor ilomastat was cardioprotective in your (and other’s) previous animal study, but not in this one. Please discuss this difference – for example, try to find differences in the two sets of experimental procedures (animals, dosages, time courses, evaluation methods etc.). If you come to the conclusion, that there are no reasonable experimental differences, you either have to correct your previous paper, or you have to perform experiments with ilomastat again to find out why you failed to show a cardioprotective effect with that MMP-inhibitor now. Otherwise, your current results are questionable, it might just be accidential fluctuations you observe with the MMP2-inhibitors. Despite statistics, reproducibility of results are obligatory – not necessarily regarding absolute values, but general outcomes.

That said, however, the infarct sizes of your control groups are quite differing. The “cardioprotective” effect of the MMPIs seems to be well within the fluctuations of the various vehicle groups (see next points).

3.) There is a stiking difference in your infarct sizes between the various control groups (vehicle/DMSO groups). Male wistar rats: 63,7%; Normochol rats: 55,6%; Hyperchol rats: 45,6%. These are only the means, and the distribution range of the individual values is rather large, but nevertheless it’s puzzling since the infarct sizes in the “sucessful” MMP2-inhibitor groups of the healthy rat model are between 53,5 and 56,6% - thus within the range of healthy controls.

What is the difference between “healthy male wistar rats” and “normo-choleresterolemic male wistar rats”, or in the experimental procedure? Age? The 12-hour-fasting and drawing of blood? Heparin dose/application? How can the infarct size difference of these two groups be explained?

4.) The difference of infarct sizes of the 2 vehicle groups in the comorbid model may be equally problematic, as shown in Fig. 3: the infarct size of the hyperchol veh group is smaller than in the normochol group, yet the infarct sizes of all MMPI-groups are about in the same +/-40% range.

Again, I recognize the valor of statistics, but can you explain the reduced infarct sizes in the hyperchol model? Does this disguise a positive effect of MMPIs? The passage in the discussion referring to the hypercholesterolemia model does not mention this.

Why are two different post-ANOVA tests (Fisher LSD and Dunnett’s MC) used?

Minor points:

Section 2.1.1: What was included in the “tolerance testing” to determine “safety” – only overall mortality, so a LD50-value or such, or behaviour, organ performances etc.? Line 108/109: You mean to say, the use of DMSO was limited to 60 µl per rat which did not affect mortality? Please rewrite.

Lines 115/116: Here, in the results section of the paper, only MMPI-1154 and -1260 were found to be cardioprotective. Please state it that way, or better leave that sentence out at this point.

Supp Figs. 6/14: What does “incidence” refer to: Number of animals, who experienced arrhythmias? Or number of episodes? Or time? Please state in graphic/heading.

Fig 3: Please write “fasting” instead of “starvation”